# Allosteric Modulation of Cannabinoid Receptor 1—Current Challenges and Future Opportunities

**DOI:** 10.3390/ijms20235874

**Published:** 2019-11-22

**Authors:** Szymon Hryhorowicz, Marta Kaczmarek-Ryś, Angelika Andrzejewska, Klaudia Staszak, Magdalena Hryhorowicz, Aleksandra Korcz, Ryszard Słomski

**Affiliations:** 1Institute of Human Genetics, Polish Academy of Sciences, Strzeszyńska 32, 60-479 Poznań, Poland; szymon.hryhorowicz@igcz.poznan.pl (S.H.);; 2Department of Structure and Function of Retrotransposons, Institute of Bioorganic Chemistry, Polish Academy of Sciences, Noskowskiego 12/14, 61-704 Poznań, Poland; 3Department of Integrative Genomics, Institute of Anthropology, Adam Mickiewicz University, Uniwersytetu Poznańskiego 6, 61-614 Poznań, Poland; 4Department of Biochemistry and Biotechnology, University of Life Sciences, Dojazd 11, 60-632 Poznań, Poland

**Keywords:** cannabinoid receptor type 1, CB1R, CB 1 receptor, GPCR, allosteric modulation, functional selectivity, biased signaling, drug development

## Abstract

The cannabinoid receptor type 1 (CB1R), a G protein-coupled receptor (GPCR), plays an essential role in the control of many physiological processes such as hunger, memory loss, gastrointestinal activity, catalepsy, fear, depression, and chronic pain. Therefore, it is an attractive target for drug discovery to manage pain, neurodegenerative disorders, obesity, and substance abuse. However, the psychoactive adverse effects, generated by CB1R activation in the brain, limit the use of the orthosteric CB1R ligands as drugs. The discovery of CB1R allosteric modulators during the last decade provided new tools to target the CB1R. Moreover, application of the site-directed mutagenesis in combination with advanced physical methods, especially X-ray crystallography and computational modeling, has opened new horizons for understanding the complexity of the structure, function, and activity of cannabinoid receptors. In this paper, we present the latest advances in research on the CB1R, its allosteric modulation and allosteric ligands, and their translational potential. We focused on structural essentials of the cannabinoid 1 receptor- ligand (drug) interactions, as well as modes of CB1R signaling regulation.

## 1. Introduction

G protein-coupled receptors (GPCRs) are integral membrane glycoproteins representing the largest family of cell-surface receptors, characterized by a presence of the seven-α-helical- transmembrane spanning (7TM) structural motif. More than 1200 GPCRs mediate responses to peptides, proteins, hormones, neurotransmitters, metabolites, ions, photons, fatty acids, pathogens, and also physical stimuli such as a sense of vision, olfaction, and taste in the human body [1,2,3,4,5]. Moreover, they play an essential role in cell activation, differentiation, proliferation, and directed migration, including immune and inflammatory response processes [6]. These receptors can mediate transduction of extracellular signals to control various physiological functions via G protein dependent or G protein-independent heterotrimeric pathways. Incorrect GPCR signaling, which may have grounds in genetic variability, acquired or inherited receptor mutations, changes in the ligand binding specificity and impaired regulation of receptor function lead to disorders in most tissues and organs of the human body [7,8].

Cannabinoid receptor type 1 (CB1R) is the most abundant GPCR, of the class A receptor, expressed mainly in neurons in the central nervous system [9,10,11]. It is one of the two best-known receptors of the cannabinoids and is an integral part of the endocannabinoid (ECS) system. The entire endocannabinoid system is comprised of (1) the endogenous cannabinoids (endo-cannabinoids), anandamide (N-arachidonoylethanolamide, AEA) and 2-arachidonoylglycerol (2-AG), which are physiological ligands for cannabinoid receptors; (2) the cannabinoid receptors (CB1R and CB2R); and (3) the enzymes responsible for synthesis and degradation of the endocannabinoids (fatty acid amide hydrolase, FAAH, or monoacylglycerol lipase, MGL). ECS plays an essential role in the proper functioning of the central and autonomic nervous system. It controls the modeling of neural connections and synaptic signaling. ECS is also involved in the regulation of emotional states, motor movement, hormonal, immune, and digestive systems [12,13]. CB1R mediates processes such as hunger, memory loss, digestive tract activity, catalepsy, fear, depression, and chronic pain [14,15,16]. Since the CB1R is involved in many physiological and pathophysiological processes, it is an attractive target for drug discovery to manage pain, neurodegenerative disorders, obesity, and substance abuse.

Over the last decade, the number of reports on the structure and function of the CB1R, its allosteric ligands, and their translational potential has increased enormously. Application of the site-directed mutagenesis together with advanced physical methods (NMR, EPR, MS, FRET, and X-ray crystallography) and computational modeling opened new horizons for understanding the complexity of the structure, function, and activity of cannabinoid receptors [17,18,19,20,21]. This review organizes the current research on CB1R allosteric modulation. We aimed to emphasize the important role of the extracellular loop ECL2 as one of the critical elements of the puzzle consisting of the cannabinoid receptor 1-ligand (drug) interactions, as well as modes of CB1R signaling regulation.

## 2. Allosteric Modulation of GPCRs

Allosteric regulation has long been recognized as a general and widespread mechanism for the control of proteins functioning as enzymes or receptors. The use of new methods and modern physical technologies in the field of receptor studies has helped to better understand this phenomenon in pharmacology and has created new directions for the development of new drugs [22]. According to Kenakin [23], GPCRs (Seven-Transmembrane Receptors—7TMRs) are Nature’s prototypical allosteric proteins and they are designed to bind multiple ligands and change their conformation accordingly to bind multiple intracellular bodies (e.g., signaling proteins) to transmit signals from the extracellular to the intracellular space.

Small molecules or proteins acting as allosteric modulators can bind to regulatory sites distinct from the active site on the protein, leading to conformational changes that may affect the function and/or activity of the protein. The International Union of Basic and Clinical Pharmacology (IUPHAR) defines an allosteric site as a “*binding site on a receptor macromolecule that is nonoverlapping and spatially distinct from but conformationally linked to, the orthosteric binding site*” [24]. The binding sites of endogenous ligands at GPCRs have been referred to as orthosteric, whereas binding sites that modulate orthosteric ligand activity have been called allosteric. Some basic definitions and concepts concerning receptors and ligands are presented in Figure 1.

Allosteric modulators bind to the receptor, leading to its distinct conformational change [25]. Such change may enhance or inhibit the activity of the receptor in the absence of an orthosteric ligand, now termed allo-agonism or allo-antagonism. The binding of the allosteric ligands may also modulate the binding affinity of orthosteric ligands, the signaling efficiency of orthosteric ligands, or may perturb signaling even in the absence of orthosteric ligands [26]. Positive allosteric modulators (PAMs) enhance receptor signaling. There are several scenarios of (PAMs) action: the receptor may have increased affinity to the orthosteric ligand or the orthosteric ligand may dissociate less efficiently. Also, the increased potency or increased efficacy of the orthosteric ligand and some combination of the above may contribute to enhanced signaling.

In contrast, negative allosteric modulators (NAMs) weaken the impact of orthosteric ligands, while neutral allosteric ligands (NALs) are compounds that bind to allosteric sites but have no functional effect on the receptor [27,28]. Schematic images of potential effect of different allosteric modulators in cooperation with orthosteric ligand on GPCR function are presented in Figure 2. Allosteric modulators have three main features that make them potentially more effective than the orthosteric ligands: a high *specificity*, target *selectivity* and *saturability* [29,30,31,32,33]. The most important is the enhanced *specificity* due to a higher level of the sequence variability in allosteric binding sites compared to conservative orthosteric domains, enabling a specific action on a given receptor subtype. The second important feature is the *selectivity* of the target action. The endogenous orthosteric ligands affect the signal pathways of the receptor in all areas where it occurs, while allosteric modulation allows for control of the receptor response only in those tissues that contain endogenous orthosteric ligand. The tissue-specific action of allosteric ligands seems to be very important for potential therapy, especially in the light of the “*synthesis on-demand*” nature of the production of endogenous ligands in the body [10]. One of the key properties is insurmountability, i.e., the ability of allosteric ligands to cause a decrease in the potency and/or efficacy of the endogenous agonist, even when the endogenous ligand is present at high concentrations.

Other advantages of the allosteric modulators are the *saturability* or ceiling effect (no further modulation is observed beyond a certain concentration of the allosteric ligand, protecting from overdosing) as well as probe dependence (the same allosteric ligand may have different effects on different orthosteric ligands) [34]. Allosteric modulators regulate the potency and efficacy of the orthosteric ligands action in a non-competitive manner [35].

A comprehensive review of allosterism concerning receptor families, including a description of methods for detection and validation of allosteric interactions, as well as recommendations for the nomenclature of allosteric ligands, was published by IUPHAR in 2014 [36].

## 3. CB1R and Its Ligands

Cannabinoid receptor type 1 (CB1R) is the most common neural receptor of the G protein-coupled receptors family [37,38]. CB1R is a product of the *CNR1* gene encoding a 53 kDa protein composed of 472 amino acids [39,40]. CB1R protein consists of seven transmembrane α-helices (TMH1–7), amphipathic helix 8, three extracellular loops (ECL1–3), and three intracellular loops (ICL1–3). The ligands interacting with CB1R can be classified according to their origin (endocannabinoids, phytocannabinoids, or synthetic cannabimimetics), chemical structure (e.g., indole, urea, or tropane derivatives) and psychoactive (psychotropic and non-psychotropic) effect [41]. Hence, CB1R can be activated by an endocannabinoid agonist such as anandamide (AEA) and 2-arachidonoylglycerol (2-AG), synthetic cannabimimetics (e.g., CP 55,940 and HU-210), or exogenous agonists like ∆-9-tetrahydrocannabinol (THC), a phytocannabinoid responsible for a psychoactive action of marihuana; synthetic inverse agonist/antagonists (e.g., SR-141716A and AM251), and neutral antagonist (e.g., AM6545) [11,15]. All of these aforementioned ligands are orthosteric ligands. Selected synthetic cannabinoids are listed in Table 1. CB1R orthosteric inverse agonists bind to the receptor at the same site as the orthosteric agonist but induce an opposite response to the agonist. In contrast, the neutral CB1R antagonist binds to the receptor at the orthosteric site but only antagonizes the endogenously released endocannabinoids [42].

For a long time, the orthosteric ligands of the CB1R were considered to be potential pharmaceuticals in the treatment of disorders such as drug addiction, obesity, and pain. However, as a result of cannabinoid receptor activation, some of those ligands caused side effects, especially adverse psychoactive effects (including depression and suicidal thoughts), which excluded them from clinical use [43]. The cannabis-based medications include Dronabinol^®^, Cesamet^®^, and Sativex^®^ [10]. Dronabinol^®^ and Cesamet^®^ are the only orthosteric modulators used in the treatment of chemotherapy-induced nausea and in AIDS patients. Because of accompanying unwanted effects, these drugs are approved by the FDA only for restricted use. Sativex^®^ is approved in some European countries for the treatment of spasticity from multiple sclerosis. Other examples of the limited therapeutic potential of CB1R synthetic orthosteric modulators are rimonabant and taranabant. Previously recommended as drugs for obesity, these compounds caused adverse effects, including depression and anxiety [14].

## 4. CB1R Activation and Signaling

Typical activation of the CB1R is based on coupling to Gi/o protein heterotrimers (i.e., α and βγ subunits), which starts a series of downstream signaling cascades that reduce intracellular cAMP concentration in most tissues and cells by inhibiting adenylyl cyclase and protein kinase A (PKA) activity [16,44,45]. Subsequently, the down-regulated PKA suppresses PKA-mediated signaling events. The dissociated βγ subunits stimulate the pathways of phosphatidylinositide 3-kinase (PI3K) and protein kinase B (PKB), which induce the phosphorylation of mitogen-activated protein kinases (MAPKs). Under certain circumstances, the CB1R can switch the G protein coupling from Gi/o to Gs or Gq. Several mitogen-activated protein kinases (MAPKs), including ERK1/2, p38, and JNK, are activated by the CB1R. The phosphoinositide 3-kinase (PI3K)/protein kinase B (Akt) pathway is activated by CB1R as well [13,44,45].

CB1R can also transduce signaling by non-G proteins pathways including the β-arrestins, the adaptor protein AP-3, GPCR-associated sorting proteins (GASP), the factor associated with neutral sphingomyelinase (FAN), and the cannabinoid receptor-interacting protein 1a (CRIP1a). These partners regulate downstream effectors including MAPKs, multiple receptor tyrosine kinases and extracellular signal-regulated kinases (ERK). Signal transmission by neurotransmitters can also be regulated by modulation of calcium and potassium channels. The CB1R can suppress calcium influx via voltage-gated calcium channels (VGCC), activate A-type VGCC, and G protein-coupled inwardly rectifying potassium channels (GIRK) [13,44].

## 5. CB1R Biased Signaling 

The concept of biased agonism of GPCR was introduced by Kenakin in 1995 [46]. Historically, it was believed that GPCR signaling was mainly transmitted through the coupling of multiple G α proteins. This view was revised after β-arrestin was detected and confirmed to be able to independently mediate diverse GPCR signaling. The molecular mechanism of β-arrestin-biased agonism was reviewed in detail by Reiter et al. [47]. Upon activation by an agonist, the 7TMR (GPCR) can selectively activate either the G protein or β-arrestin pathway, which is called “biased signaling”, “ligand bias”, “biased agonism” or “functional selectivity”. The term “biased agonism” means that GPCRs may be biased toward, either G protein-dependent pathway or β-arrestin-dependent pathway, and ligands that selectively activate either G protein or β-arrestin-dependent pathways are referred to as “biased ligands”. However, most of the endogenous ligands of GPCRs are capable of activating both of these pathways. Simplified scheme of classical and biased signaling is presented in Figure 3.

Biased agonism also can be described as a phenomenon whereby a ligand can be an agonist for one pathway downstream of the receptor and at the same time is either a neutral antagonist or an inverse agonist for another downstream signaling pathway(s) [48]. An example is the CB1R allosteric modulator Org27569, which is a positive allosteric modulator of CP 55,940 (orthosteric agonist) affinity but a negative allosteric modulator of CP 55,940 efficacy for mediating CB1R-dependent cAMP inhibition [49].

According to Reiter et al. [47], there are several important consequences of this phenomenon such as pluridimensional nature of pharmacological efficacy, and the possibility of creating multiple ligand-specific conformations by activated 7TMRs. On the other hand, biased ligands are capable of stabilizing only a subset of the receptor conformations triggered by a balanced ligand, which enables selective activation of intracellular signaling pathways [47].

G protein or β-arrestin ligands have been identified for a large number of 7TMRs (including CB1R) and, therefore, may be considered general classes of pharmacological compounds [47]. Since “biased ligands” can selectively activate either G protein-biased or β-arrestin-biased signaling, they can activate beneficial effects and block unwanted effects induced by GPCR activation, which constitutes a substantial therapeutic potential [15].

Interestingly, there is also a possibility of CB1R signaling through both G proteins and β-arrestin pathways. Noguera-Ortiz and Yugowski [50] postulated that the activation of CB1R can change over time and occurs in three different waves. The first wave is transient and involves heterotrimeric G proteins (Gα, β, γ). The second wave is mediated by β-arrestins. Finally, the third wave takes place in the intracellular compartments and can occur either by G proteins or β-arrestins.

## 6. Other Novel Modes of CB1R Activation

There are four other novel modes of GPCR activation: oligomerization (homodimers of receptor or heterodimers with other receptors), intracellular activation (activation inside the cell); transactivation, and biphasic activation [27,51,52]. CB1Rs can form both homomeric and heteromeric complexes with other GPCRs, which can be considered a form of allosteric modulation [29,53]. CB1Rs can dimerize with adenosine A2A, dopamine D2, orexin, β-adrenergic, and µ-, κ- and δ- opioid receptors [29]. Callen et al. [54] showed that CB1Rs and CB2Rs can form functional heteromers in the brain. These CB1R-CB2R heteromers, expressed in a neuronal cell model, showed agonist co-activation of CB1Rs and CB2Rs resulting in negative cross-talk in Akt phosphorylation and a bidirectional cross-antagonism phenomenon. Sierra et al. [55] detected CB1R and delta-opioid receptor heteromers, in both a mouse model and postmortem tissues of patients with the chemotherapy-induced peripheral neuropathy, which could serve as a potential target for neuropathic pain treatment.

Although most GPCRs have ligand binding sites in the extracellular domain, the existence of intracellular ligand binding sites has been suggested for chemokine receptors (CCRs). The recent X-ray structures of CCR2, CCR9, and β-2AR in complex with an orthosteric antagonist and inhibitor have revealed a highly-conserved intracellular pocket for small molecules, suggesting its presence in most CCRs and other members of class A GPCRs [56]. Intracellular activation of CB1R was detected in lysosomes and mitochondria [52,57]. CB1R was localized in the outer membrane of neuronal mitochondria and found to regulate neuronal energy metabolism [57]. Further studies in mice demonstrated that upon activation by exogenous cannabinoids and in situ endocannabinoids, mitochondrial cannabinoid receptor (“mtCB1”) induced an intra-mitochondrial signaling pathway involving G proteins, soluble adenylyl cyclase (sAC), and PKA, leading to reduction of complex I enzymatic activity and respiration in neuronal mitochondria [58]. Hebert-Chatelain et al. [58] showed that the G protein-coupled mtCB1 receptors regulate memory processes via the modulation of mitochondrial energy metabolism, which means that bioenergetic processes are primary acute regulators of cognitive functions. These new CB1R modes of activation point to the complexity of regulating those receptors that control so many physiological and pathophysiological processes, while providing new tools for the development of new drugs.

## 7. A Glimpse into CB1R Structure: The Second Extracellular Loop (ECL2) as a Significant Region of the CB1 Receptor

The GPCR extracellular (EC) region is one of the most active segments during receptor biogenesis. The extracellular loops (ECL) can control ligand trafficking into TM (transmembrane) bundles, change the shape of the binding pocket or bind directly to orthosteric or allosteric ligands, and play a critical role in GPCRs functioning [59]. GPCR crystal structures and functional studies have shown that ECL2 (connecting TMH4 and TMH5) is notably important [60]. Significance of the ECL2 as a key receptor region has been shown for: V_1a_ vasopressin receptor [61], rhodopsin [62], M_2_ acetylcholine receptor [63], muscarinic receptor [64], adenosine A1 receptor [65], β-1 adrenergic or β-2 adrenergic receptor [64,66], and cannabinoid receptor 1 [67,68].

The second extracellular loop (ECL2) in the class A GPCRs is the largest and most divergent of all three ECLs. It can differ (even within subfamilies) in length and sequence [64]. Based on two large phylogenetic studies [69,70], GPCRs were divided into 19 subgroups (A1–A19) or five main families including rhodopsin (α-, β-, γ-, δ-groups). CB1R was assigned to subgroup A13 next to subgroup A17 with AA_2A_R and βARs and these receptors altogether belonged to a larger cluster different from subgroup A16 (rhodopsin) [69]. In the second study, CB1R was assigned to MECA cluster (melanocortin, endothelial differentiation, cannabinoid, and adenosine binding) and revealed that the closest receptors to cannabinoid receptors are the melanocortin receptors [70].

The second extracellular loop in CB1R, which covers an amino acid sequence of 21 residues from tryptophan W255 to isoleucine I271, stabilizes the receptor structure by linking transmembrane domains TMH4 and TMH5. It also plays a significant role in ligand binding and receptor activation. ECL2 in CB1R is shorter than, belonging to the same class A, rhodopsin, and does not have a conserved disulfide bridge between the cysteines in ECl2 and C3.25 in TMH3 (of rhodopsin), that occurs in a majority of class A GPCRs and is very important for the stability of receptor structure. This is due to the lack of cysteine at the N-terminal end of TMH3 in CB1R [71]. However, in ECL2 of the CB1R, there is an intra-loop disulfide bond between Cys257 and Cys264 that is critical for receptor stability [71,72]. It was shown that mutation of these two ECL2 Cys residues to alanine abolished the binding activity of SR141716A most likely due to the breaking of the disulfide bond, significant conformational change, and modification of the ligand binding pocket [71].

ECL2 is a major initiator of allosteric communication in the ligand-based community network in CB1R. Previous studies have shown that the four residues (F268, P269, H270 and I271) are crucial for receptor biogenesis, allosteric communication, and mediating interactions with certain classes of ligands, as well as two cysteines (Cys257 and Cys264), which are required for high-level expression and receptor function [67,71,73]. It is well known that CB1R extracellular domains play an important role in ligand binding, however, Ahn et al. [67] showed that the C-terminal region of the ECL2 loop distinguishes agonist binding from inverse agonist/antagonist binding. Moreover, molecular modeling of the receptor suggests that four crucial residues (268–271), mentioned earlier, of ECl2 can project into ligand binding pocket and are especially important for accommodating both the bicyclic or tricyclic core and the alkyl side chain of cannabinoids. Furthermore, the mutation (substitutions) of F268W and I271A show greater affinity for ligand binding including some inverse agonists/antagonists, which is consistent with the fact that both of these residues are located deep in the core and form part of the ligand binding pocket [67,73]. In Figure 4, the snake diagram of human CB1R is presented; the conserved residues are highlighted in red circles.

Among all of the component loops, ECL2 is the biggest and most structurally differentiated region within the various GPCRs subfamilies, suggesting its functional importance [60]. The low level of interspecies variability of this region may also indicate its functional significance. Comparison of the human *CNR1* nucleotide gene sequence with phylogenetically distant species, *Xenopus tropicalis* and *Danio rerio*, show a significant percentage of highly conserved sequences within ECL2, despite low coverage of the whole gene sequence (Table 2).

A comparison of the ECL2 amino acid sequence in different species also showed a high percentage of conserved sequences within this region (Figure 5, Table 2). The first two amino acids of the ECL2 domain (W255 and N256) might have a crucial effect on the CB1R protein transport from the Golgi apparatus, its location in a membrane and regular activity, whereas two cysteine molecules (C257 and C264) create a disulfide bond within the ECL2 domain a and are essential for high-level expression and interaction with ligands [71,72].

The CB1R N-terminus involved in the biosynthesis and orientation of the receptor is relatively long and has in its structure a highly conserved membrane-proximal region (MPR), which affects the receptors’ ability to bind ligands through two cysteine residues (C98 and C107), forming a disulfide bond within this domain. It was suggested that the ECL2 domain and the N-terminus MPR act concomitantly, influencing the binding site of the orthosteric ligand. It was also hypothesized that it may also be the region of action of the allosteric modulators [74].

There are some other structural properties unique to CB1R. Most of the class A GPCRs are characterized by a presence of CWxP structural motif (including residue W6.48) located in their binding pocket that works as a “toggle switch” for receptor activation upon agonist binding. However, the crucial aromatic residue at 6.52 position located in TMH6 is missing both in CB1R and CB2R. In mutation studies, it was shown that the W258 (6.48) could pair with the F117 (3.36) residue to form a “toggle switch” in the CB1R, with consequent loss of aromatic stacking, leading to receptor activation [72]. Similarly to other GPCRs, CB1R features the highly conserved aspartic acid-arginine-tyrosine (“DRY”) motif, which plays an essential role in regulating both GPCR conformation and activity. Double mutations (D213 (3.49) and R214 (3.50)) in “DRY” motif caused CB1R to bias towards β-arrestin signaling and away from G protein activation, like in other GPCRs, without significant loss in binding affinity of cannabinoid agonists [75]. Gyombolai et al. [75] also showed that C3556.47 of the CWxP motifis essential for the binding of CP 55,940 and other classical cannabinoids known to induce receptor internalization through β-arrestin-mediated pathways.

In 2016 the first CB1R X-ray crystallography structures were generated when two research groups independently elucidated the first crystal structure of the CB1R. Hua et al. [18] determined a crystal structure of human CB1R in complex with the designed antagonist, AM6538, whereas Shao et al. [17] generated crystal structure of CB1R by binding the anti-obesity drug taranabant (Table 1). Despite some discrepancies and differences in resolution, both groups discovered a binding pocket that allows lipophilic cannabinoid molecules to interact with the receptor. Crystallographic studies confirmed the role of the ECL2 region in the process of ligand binding, and in effect, activation of the CB1R in cooperation with N-terminus and contribution of transmembrane domains [17,18,19]. Common structural features between the GPCRs classes were also observed. Structural similarities define common mechanisms of the ligand-binding, and sequence variability ensures the specificity of these bonds. In structural studies using the cryo-EM method, CB1R bound to Gi protein (as CB1R-Gi signaling complex) and synthetic highly potent agonist, MDMB-Fubinaca, was analyzed by Kumar et al. [21]. It was found that Fubinaca pocket is buried in the TMH region and is capped by ECL2, which folds into the pocket with F268, which have direct hydrophobic contact with the ligand. Analysis of these results suggests that ECL2 may simplify a passage of ligand to the TMH core or ECL2 may directly act as a surface of the ligand-binding pocket [21].

## 8. Allosteric Modulators of CB1R

According to structural features, the allosteric modulators of CB1R described so far, can be classified as: indole derivatives (e.g., Org27569, Org29647, and Org27759), urea derivatives (e.g., PSNCBAM-1), endogenous ligands (lipoxin A4 and pregnenolone), and other compounds like synthetic cannabinoids (e.g., JHW007; a synthetic cannabidiol (CBD) or RTI-371, a tropane derivative) [43].

The first report concerning the identification of the allosteric site of CB1R was published in 2005 by Price et al. [49]. Scientists from Organon International characterized three allosteric modulators of CB1R: Org27569, Org29647, and Org27759, all the indole derivatives. In the beginning, the best characterized indole, Org27569, was considered a positive allosteric modulator that enhances the binding of the agonist. However, further studies showed that Org27569 acts as a negative allosteric modulator that impairs agonist binding. These controversial results helped to develop advanced pharmacological assays that showed a much more complex process of Org27569 action than was initially assumed: the allosteric action of this ligand on CB1R is time- and cell signaling pathway-dependent [76].

So far, Org27569 remains the best characterized allosteric modulator of CB1R. Further studies, focused on the functional relationship “structure-activity” (SAR) of the Org27569 modulator, are warranted. Ahn et al. [67] showed that interaction of the allosteric ligand Org27569 with the ECL2 region, specifically phenylalanine F268 (Figure 1), impairs the action of synthetic cannabinoids. Then, Marcu et al. [77] demonstrated that interaction between the TMH3 and TMH6 domains in the TMH3–4–5–6 microdomain of the murine CB1R was important in the stabilization of its inactive status. Disruption of the TMH3–6 interaction causes, in effect, a specific “start” of the receptor and signal transduction in effect. Org27569, as an inverse agonist, promotes intermediate conformation of CB1R, proving its ability to enhance the binding equilibrium of the orthosteric ligand, CP 55,940 (synthetic cannabinoid, similar in action to THC). Org27569 enhances the affinity of CP 55,940 and reduces its efficacy at the same time [77]; it may also control the CP 55,940 activity by blocking the movement of the ECL2 loop and arresting the TMH6 domain, which is important for receptor activation. Org27569 may also suppress an essential electrostatic interaction between residues of the aspartic acid D176 and lysine K373, which intensifies the G protein–directed signal transmission. As a result of the blocking interaction of Org27569, signal transduction is reduced [78].

In 2007, Horswill et al. reported PSNCBAM-1, a urea derivative chemical compound, as a potential allosteric modulator of CB1R [79]. They found that this compound had a pharmacological profile similar to that of Org27569. Both compounds increase the binding of CB1R agonists, and simultaneously inhibit their functional response. The in vivo studies in rats have shown that PSNCBAM-1 reduces food intake and body weight, suggesting the contribution of this allosteric ligand to CB1R activity control. This was the first report concerning the pharmacologically active allosteric modulator of CB1R, which in the future, may be beneficial in development of the obesity pharmacotherapy. Recently, PSNCBAM-1 and its analogs have been the subject of intensive research. SAR studies have shown that specific substitutions in the compound’s structure may modify the activity and efficacy of modulators of CB1Rs [80,81,82].

The compounds of endogenous origin, displaying allosteric features toward the CB1R receptor became a subject of interest, too. One of the endogenous molecules characterized by such activity is lipoxin 4 (LXA4; an oxygenated derivative of arachidonic acid) [83]. Lipoxin 4 acts as an anti-inflammatory agent (mediator) and is involved in the regulation of the immune system, but its activity concerning the central nervous system is yet to be elucidated. It is considered that lipoxin acts as a PAM of CB1R by strengthening the binding and activity of anandamide and [3H]-CP 55,940 (a synthetic cannabinoid). Therapeutic application of LXA4 as an allosteric enhancer of CB1R activity in an in vivo model of the β-amyloid-induced spatial memory disorder was confirmed. Other research groups presented activity study results demonstrating that LXA4 is a negative allosteric modulator (NAM) [43,84].

Other compounds that can be used as allosteric modulation include CBD, JHW007, and RTI-371. It was shown, that in the presence of Δ9-THC and 2-AG, CBD acts as a non-competitive negative allosteric modulator of CB1R. Establishing CBD as a negative allosteric modulator of CB1R may be useful in an elaboration of the selective modulatory substances of the CB1R or in combination with drugs [43,85]. Navarro et al. [86] showed that RTI-371 and selective inhibitors of DAT (dopamine active transporter), which have similar pharmacological profiles in in vivo and in vitro studies, are positive allosteric modulators of the human CB1R. It was also found that ligands RTI-370, RTI-371, and JHW007 have positive allosteric characteristics, especially in the presence of CP 55,940, the orthosteric CB1R ligand [86,87]. One of the most promising CB1R allosteric modulators was recently developed GAT100, a negative allosteric modulator that functions in the presence of the orthosteric agonist, CP 55,940, and the endocannabinoids, 2-AG, and AEA, for *β*-AR1 recruitment. The results suggest that GAT100 is stronger and more effective than Org27569 or PSNCBAM-1. The amino acids involved in binding of GAT100 are F268, P269 (both in ECL2 domain) and of crucial importance in the ligand-binding motif is the cysteine C382 [15]. An important role is also attributed to GAT211, in contrast to a PAM, that binds to an allosteric site and increases the affinity of the endogenous ligand for the orthosteric binding site [88]. What’s more interesting, the pharmacological profiling revealed that the GAT211 exhibited a strong CB1R PAM activity, which was attributable to the S-(-)-enantiomer (GAT229), and CB1R partial agonist activity, which was attributable to the R-(+)-enantiomer (GAT228), which suggests broad therapeutic potential [89].

## 9. Challenges and Perspectives for Translational Use of CB1R Allosteric Modulators

Cannabinoid receptor type 1 (CB1R) is considered as a key drug target for many diseases due to its major role in the control of both physiological and pathological processes related to metabolism, cognitive function, or pain sensation. Since the psychoactive side effects generated by activation of CB1Rs on-target in the brain limit the use of orthosteric CB1R ligands as drugs, generation of the first allosteric modulators of CB1R in 2005 has started a new era in the pharmacology of cannabinoid receptors.

The phenomena of allosteric modulation, bias signaling, and oligomerization of GPCRs provide powerful potential for the development of novel specific drugs to target CB1R. Allosteric modulators do not have intrinsic efficacy, but instead enhance or decrease the receptor’s response to orthosteric ligands, therefore, they allow for the modulation of cannabinoid receptor signaling without the desensitization, tolerance, and dependence [31,43]. Allosteric ligands have numerous advantages over orthosteric ligands such as higher receptor type selectivity, probe dependence, and no alteration of functional activity of endogenous cannabinoids. Bias signaling or functional selectivity allows for activation of selective signaling pathways and downstream responses by allosteric modulators, resulting in generation of unique and therapeutically beneficial patterns of signal transduction pathways [90,91]. Traditionally, biased signaling and allosteric modulation of GPCRs by ligands were considered to be separate phenomena, but both are due to ligand-specific conformational changes in the GPCR that involve a change in the “shape” of the receptor [27]. Analysis of the vast amount of data based on numerous structure-function and pharmacological studies has led to the conclusion that GPCRs work as allosteric microprocessors rather than as binary “switches” [92].

Oligomerization has great potential and is a challenge since this feature affects the affinity and specificity of ligand binding, the pattern of signaling, and internalization [27]. Ligand bias can be conditioned by receptor oligomerization in specific cells or tissues as CB1R-dependent Gα_q_ signaling was shown to occur through CB1R-D2R (dopamine receptor type 2) dimerization [93].

The promising perspective is the development of drugs designed to accurately modulate the ECS. The ligands that have opposing functional profiles (CB2R antagonist and CB1R agonist) support the thesis of the yin-yang functional relationship of these two receptors and the possibility of elimination of neuropsychiatric side effects [20,94].

Allosteric modulation is a very complex process and became even more complicated with the introduction of bivalent and multifunctional ligands that can serve as advanced pharmacological tools to characterize homo- and hetero-dimerization of GPCRs and physiological interactions between receptor systems [95]. Multifunctional (or multi-target directed) ligands (MTDLs) targeting cannabinoid and other receptors, ion channels, or enzymes with multiple pharmacological activities might display therapeutic advantages in the treatment of multifactorial disorders or diseases.

Despite extensive studies on the two best known allosteric modulators, Org27569 and PSNCBAM-1, they are not fully profiled in vivo and a major limitation related to them is that they exhibit CB1R inverse agonism in addition to NAM activity [96]. Therefore, efforts should be directed towards identifying of potent and efficacious CB1R NAMs, which are free of inverse agonism, as well as those exhibiting functional selectivity with drug-like physicochemical properties, which would resolve the beneficial versus deleterious effects of CB1R activation, thereby reducing the side effects of modulating/attenuating CB1R signaling [96].

One of the major challenges is translating in vitro data to in vivo effects. The differences are due to interspecies variation, e.g., compounds that are identified as promising based on the screens in human cell lines do not produce the desired effect when tested in an appropriate animal model [27]. Cell type differences also relate to differences when recombinant versus a natural cell system is used or when different recombinant systems, utilizing different host cells, are compared [23].

The greatest challenge and goal of studying bias signaling for therapy is the ability to predict, identify, and correlate a specific signaling cascade and associated effector proteins with behavioral and/or pathophysiological outcomes in vivo [76,97].

The strategies to overcome this challenge should be intensified in several directions: further structural studies to characterize receptor in complex with ligand in active and inactive states; development of more pharmacological tools/assays to better quantitate kinetics of ligand binding and bias signaling; and more accurate characterization of ligand binding sites through radioligand binding, mutagenesis, and modeling studies.

## 10. Conclusions

Allostery with biased signaling and dimerization is very complex and researchers are at the beginning stages of their understanding of this signaling interaction; but with much effort, and insight, the nuisances of allostery can be elucidated, which will be fruitful in the development of novel, safe, and efficacious drugs with no neuropsychiatric side effects.

## Figures and Tables

**Figure 1 ijms-20-05874-f001:**
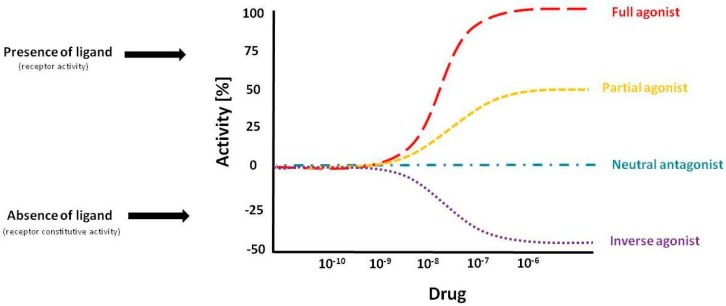
Basic definitions and concepts concerning a receptor and its ligand in pharmacology. A ligand that binds and activates the receptor is called an agonist. Full agonism means a maximum receptor activation, resulting in maximum system response. Partial agonist produces submaximal receptor activation, leading to the production of submaximal system response and possible blockade of full agonist activation. A ligand that binds to a receptor, but does not activate it, or inactivate it, is called an antagonist. Antagonists produce their effects by preventing agonists from binding to the receptor. Antagonists produce no physiological response but rather block the response to endogenous or exogenous agonists. Inverse agonism can be seen when a receptor has high basal activity in the absence of agonist (constitutive activity). Ligand that binds to a receptor and inactivates it, resulting in decreased signaling, is called an inverse agonist. Inverse agonist functions as an antagonist in non-constitutively active systems, but has the added property of actively reducing receptor-mediated constitutive activity of GPCR systems (response not resulting from agonist activation but originated from the system itself). We can also distinguish allosteric agonist, which activates the receptor via interaction at a different site than the endogenous agonist; an allosteric modulator antagonist, which blocks the function of the receptor or interferes with the ligand-receptor interaction; and an allosteric enhancer that enhances the action of agonists [5,25]. Three parameters describe allosteric modulators more quantitatively: Affinity is a parameter describing how strongly a ligand binds to its target; Efficacy describes the degree of effect or response achieved by a specific ligand upon binding to its target; Potency is a parameter describing the activity of a drug by defining how much ligand (concentration) is needed to produce a half-maximal effect.

**Figure 2 ijms-20-05874-f002:**
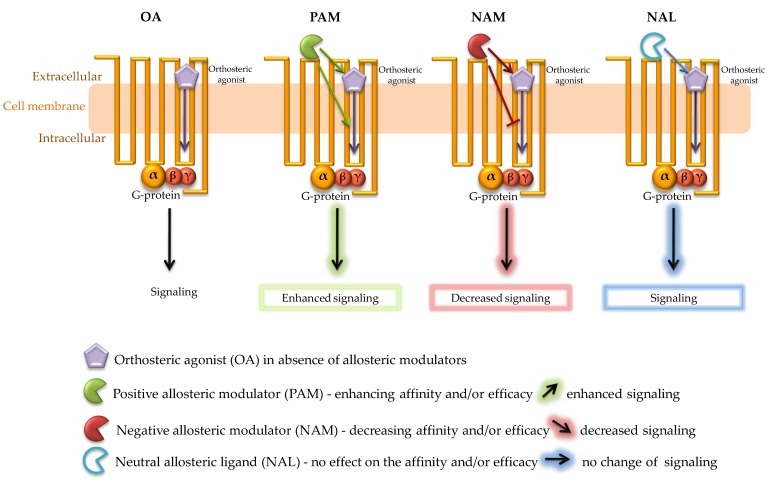
Schematic presentation of the effect of different GPCR allosteric modulators on the orthosteric ligand function. Orthosteric agonist (**OA**) binds to GPCR leading to a conformation change that activates a signal cascade. In the presence of (**OA**) upon binding of (**PAM**) to the allosteric site of GPCR, a change in receptor conformation leads to enhanced receptor signaling through increased potency, affinity and/or efficacy of the orthosteric agonist. (**NAM**) works opposite to PAM, i.e., it reduces the affinity and/or efficacy of the orthosteric agonist at the receptor and leads to a decrease in receptor signaling. In the case of (**NAL**), no change in signaling is observed.

**Figure 3 ijms-20-05874-f003:**
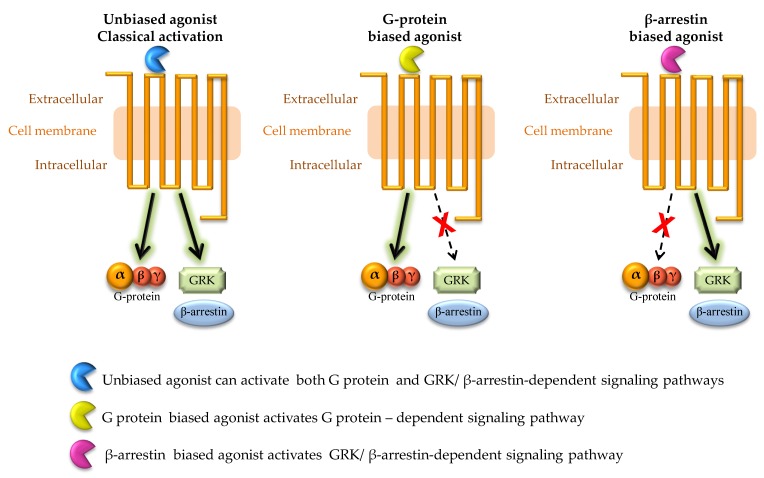
Simplified scheme representing classical and biased GPCR signaling. In classical (unbiased) GPCR activation, a G protein-dependent as well as β-arrestin-dependent pathways can be activated. In case of binding of biased agonist, the signaling can be biased either toward G protein-dependent signaling pathway or β-arrestin-dependent signaling pathway.

**Figure 4 ijms-20-05874-f004:**
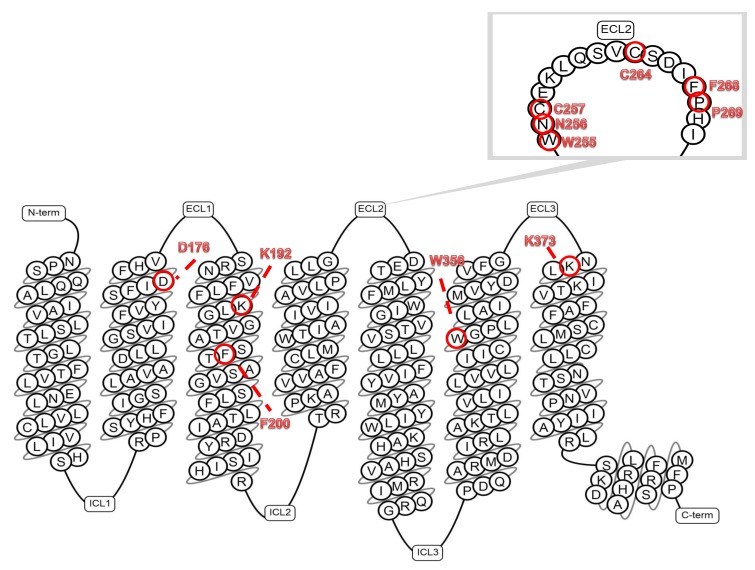
Snake diagram of the human CB1R. CB1R structure is typical for GPCR and contains three extracellular loops (**ECL1–3**), three intracellular loops (**ICL1–3**), seven transmembrane domains (TMH1–7), and a C-terminus. The amino acid sequence of the (TMH1–7), C-terminus and the (**ECL2)** (zoomed) domains are shown. The amino acids involved in the receptor-ligand interactions are marked in bold red font. Red circle labels indicate amino acids involved in interactions of the allosteric ligand Org27569 with the (**ECL2**) region, specifically phenylalanine F268 and lysine K192 (TMH3) and also with aspartic acid D176 (TMH2) and lysine K373 (TMH7). Phenylalanine F268 and proline P269 (**ECL2**) are also essential for binding an allosteric ligand GAT100. (Generated by tools from gpcrdb.org).

**Figure 5 ijms-20-05874-f005:**
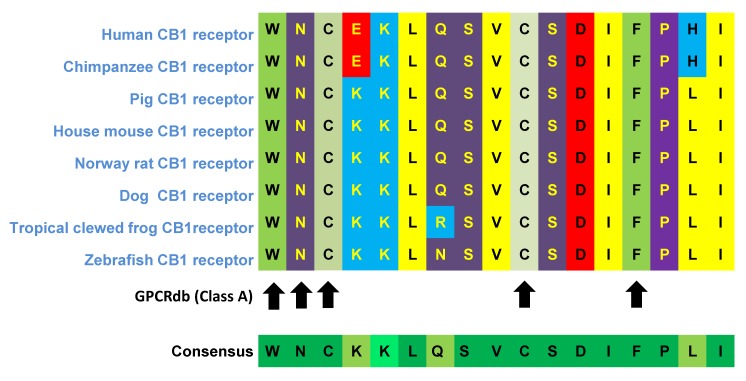
Sequence alignment of the ECL2 region of CB1R in 8 different species (Generated by tools from gpcrdb.org). The ECL2 sequences presented cover 17 amino acids from tryptophan W255 to isoleucine I271. The arrows point to the most important conserved residues: tryptophan W255; two cysteines C257 and C264 forming disulfide bridge; phenylalanine F268 involved in ligand interaction.

**Table 1 ijms-20-05874-t001:** Selected synthetic cannabinoids.

Compound	IUPAC Nomenclature
CP 55,940	[2-((1*R*,2*R*,5*R*)-5-Hydroxy-2-(3-hydroxypropyl) cyclohexyl)-5-(2-methyloctan-2-yl)phenol]
HU-210	[(6a*R*,10a*R*)-9-(Hydroxymethyl)-6,6-dimethyl-3-(2-methyloctan-2-yl)-6a,7,10,10a-tetrahydro-6*H*-benzo [c]chromen-1-ol]
WIN 55212-2	[(*R*)-(5-Methyl-3-(morpholinomethyl)-2,3-dihydro-[1,4]oxazino [2,3,4-hi]indol-6-yl)(naphthalen-1-yl)methanone]
JWH-018	[Naphthalen-1-yl-(1-pentylindol-3-yl)methanone]
SR-141716A (rimonabant)	[5-(4-Chlorophenyl)-1-(2,4-dichlorophenyl)-4-methyl-N-(1-piperidinyl)-1*H*-pyrazole-3-carboxamide]
AM251	[1-(2,4-Dichlorophenyl)-5-(4-iodophenyl)-4-methyl-N-(1-piperidinyl)-1*H*-pyrazole-3-carboxamide]

**Table 2 ijms-20-05874-t002:** Comparison of the human *CNR1* nucleotide gene sequence and extracellular loop 2 (ECL2) (Generated by Blast and M7 alignment explorer).

Human Homo Sapiens (NC_000006.12) vs	*CNR1* Gene		ECL2 Region
	Coverage	Identity	Identity
Chimpanzee Pan troglodytes (NC-006473.4)	99%	99%	99%
Pig Sus scrofa (NC_010443.5)	56%	85%	88%
House mouse Mus musculus (NC_000070.6)	29%	83%	92%
Norway rat Rattus norvegicus(NC_005104.4)	28%	83%	92%
Dog Canis lupus familiaris(NC_030681.1)	50%	84%	92%
Tropical clawed frog Xenopus tropicalis (NC_030681.1)	4%	76%	78%
Zebrafish Danio rerio (NC_007131.7)	3%	76%	72%

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
