# Peer review of "Allosteric Modulation of Cannabinoid Receptor 1—Current Challenges and Future Opportunities"

_ijms, 2019, doi:10.3390/ijms20235874_

Round 1
Reviewer 1 Report
In this review, the authors present the latest advances in research on the cannabinoid receptor 1 (CB1R), its allosteric modulation and allosteric ligands, and their translational potential. They focused on structural essentials of the cannabinoid 1 receptor-ligand (drug) interactions, as well as modes of CB1R signaling regulation.
This, comprehensive review of allosteric modulators of the CB1R and their potential as therapeutic agents, is interesting and well-thought.

Author Response
Thank you very much for reading and evaluating our manuscript.
We very much appreciate all the comments, amendments, and especially the addition of own sentences to the conclusions of our manuscript by Reviewer 1. We are very grateful for this refined contribution, which significantly improved our manuscript.
We introduced all the corrections (red font on yellow background) suggested by Reviewer 1. We added two additional figures (Fig. 2 and Fig.3) to the manuscript according to suggestions of Reviewer 2.
Additionally, we changed the text labelled by yellow background in following paragraphs:
Lines 114-117 (p.3); lines 232-236 (p.7); lines 366-369 (p.10) and lines 375-381 (p.10). The text was changed due to suggestions of the editorial office because of 80% repeated parts in these fragments compared with contents in other materials.
Here, we attach the corrected manuscript file as Word format with “Track changes” function active.

Reviewer 2 Report
The review of Hryhorowicz et al. is a very well planned and edited paper about an interesting hot topic, the usefullness of AM for the CB1 receptor subtype.
It gives exhaustive piece of information about the perspectives of this type of ligands for the target receptor.
Maybe it could be useful to the reader to have some figures showing how these ligands work in a schematic way.
Author Response
Thank you very much for reading and evaluating our manuscript.
According to suggestions of Reviewer 2, we added two additional figures (Fig. 2 and Fig.3) to the manuscript to illustrate better the action of allosteric modulators and the basic example of biased signaling.
Additionally, we changed the text labelled by yellow background in following paragraphs:
Lines 114-117 (p.3); lines 232-236 (p.7); lines 366-369 (p.10) and lines 375-381 (p.10). The text was changed due to suggestions of the editorial office because of 80% repeated parts in these fragments compared with contents in other materials.
Here, we attach the corrected manuscript file as Word format with “Track changes” function active.
